# Automatic Pest Monitoring Systems in Apple Production under Changing Climatic Conditions

Dana Čirjak [1,*], Ivana Miklečić [1], Darija Lemić [1], Tomislav Kos [2] and Ivana Pajač Živković [1]

1   Department of Agricultural Zoology, Faculty of Agriculture, University of Zagreb, Svetošimunska cesta 25, 10 000 Zagreb, Croatia; imiklecic@agr.hr (I.M.); dlemic@agr.hr (D.L.); ipajac@agr.hr (I.P.Ž.)
2   Department of Ecology, Agriculture and Aquaculture, University of Zadar, Trg kneza Višeslava 9, 23 000 Zadar, Croatia; tkos@unizd.hr
*   Correspondence: dcirjak@agr.hr; Tel.: +385-1239-3948

**Abstract:** Apple is one of the most important economic fruit crops in the world. Despite all the strategies of integrated pest management (IPM), insecticides are still frequently used in its cultivation. In addition, pest phenology is extremely influenced by changing climatic conditions. The frequent spread of invasive species, unexpected pest outbreaks, and the development of additional generations are some of the problems posed by climate change. The adopted strategies of IPM therefore need to be changed as do the current monitoring techniques, which are increasingly unreliable and outdated. The need for more sophisticated, accurate, and efficient monitoring techniques is leading to increasing development of automated pest monitoring systems. In this paper, we summarize the automatic methods (image analysis systems, smart traps, sensors, decision support systems, etc.) used to monitor the major pest in apple production (*Cydia pomonella* L.) and other important apple pests (*Leucoptera maifoliella* Costa, *Grapholita molesta* Busck, *Halyomorpha halys* Stål, and fruit flies—Tephritidae and Drosophilidae) to improve sustainable pest management under frequently changing climatic conditions.

**Keywords:** apple pests; automated pest monitoring; climate change; *Malus domestica* Borkh; precision agriculture



## 1. Introduction

Although the scientific development of genetics, chemistry, and robotics has contributed to the advancement of agricultural technology, various problems still arise. For example, the production of agricultural goods must be increased due to rapid population growth [1]. The availability of arable land and fresh water is becoming extremely limited [2]. Unsustainable agriculture threatens crop productivity and the environment in addition to causing anomalies in the climate. These types of practices also create difficulties in the relationship between production and consumption [3]. At the same time, climate change problems are becoming more severe and affecting many aspects of agricultural production. Changes in yields, harvest times, farming practices, pest incidence, and many other crucial factors are directly influenced by changing climatic conditions [4–7].

A suitable solution to address the above situations and challenges can be found in artificial intelligence (AI) technologies, which are helping to improve efficiency in many sectors, including agricultural [8]. Crop yields, irrigation, soil content recording, crop monitoring, weeding, pest monitoring, and crop establishment have recently been accompanied by AI in the context of precision agriculture (PA) [2,9–11]. Precision agriculture can be defined as the use of technologies and principles to manage all aspects of agricultural production to improve crop yield and preserve the environment [12].

This, along with other important approaches, represents a critical agricultural management system that combines the use of robotics and sensors, drones, advanced GPS (Global Positioning System) and GNSS (Global Navigation Satellite Systems), IoT (Internet of

Things), weather modeling, and the tailored use of inputs [3,13]. It is a cyclic optimization process in which data from the field are collected, analyzed, evaluated, and finally used to make decisions for site-specific management of the field.

With such a working principle, these systems allow farmers to analyze the spatiotemporal variability of several key factors that affect crop health and productivity [14]. Data collected via sensors are stored and combined in digital platforms to support the decision-making process. Ideally, the farmer should be able to maximize yields while optimizing inputs, saving nutrients, and replacing labor time with efficient decision support systems, which can increase farm profitability and reduce the dependence on external inputs and thus negative environmental impacts [3,14].

In cultivation, environmental factors are key to crop quality and productivity. Among these factors, insect pests are those that directly damage crops, and pest control has always been considered the most difficult challenge. Therefore, integrated pest management (IPM) has been developed to improve pest control, reduce the uncontrolled use of pesticides, and focus on more precise application [15]. The effectiveness of pest control programs depends on the availability of reliable and up-to-date pest infestations information. Intervention thresholds derived from catches in monitoring traps are a cornerstone of modern IPM programs to trigger and optimize the timing and use of insecticide sprays.

IPM, however, requires intensive field observation, trained personnel, and data evaluation. Weekly trap inspections and close observations of plants in the field can lead to delayed intervention and involve some labor. Without collecting information on population dynamics and associated ecological factors, it is extremely difficult to apply the right pest control at the right time in the right place.

Moreover, pest harmfulness is associated with plant physiology; thus, early pest detection during critical phenophases of development is necessary in order to implement control measures in a timely manner and prevent the increase of the pest population and damage on fruits, which is the key aspect of IPM. Therefore, to improve the efficiency of data collection and to perform more accurate and reliable pest control, it is necessary to use automated monitoring systems [15–20]. To effectively control and prevent the occurrence of pests, many advanced technological solutions have been developed and applied in today's agricultural and crop industries [21–23].

Apple (Malus domestica Borkh.) (Rosales: Rosaceae) is one of the most important fruit crops worldwide. Since it can be used for fresh consumption all year round or processed into a product, such as apple sauce, apple slices, ciders, and juice [24,25], global apple production in 2020 amounted to 86.4 million tons, with an economic value of US$ 77 billion, while the apple area harvested was 4.62 million ha [26]. Due to its aforementioned economic importance, as well as the harmfulness of its pests, which significantly impair the production profitability, automatic pest monitoring systems are presented in this work on the example of apple crops.

The aforementioned problems have major implications for pest monitoring, their control, and modern agriculture in general. An acceptable solution for pest monitoring under changing climatic conditions is automatic pest monitoring systems. Therefore, this paper aims to summarize the automatic devices and techniques that can currently be used in apple production to monitor important economic pests and to assess the commercially available devices and techniques and their impact on sustainable fruit production.

## 2. Impact of Changing Climatic Conditions on Pest Monitoring

Scientific studies in agricultural science have recently focused on climate change and all the events that accompany it [7]. Climate change can be described as a phenomenon that involves variations in environmental factors, such as temperature, humidity, and precipitation, over a long period of time [27]. The most common problems caused by climate change are the increase in global temperature and atmospheric carbon dioxide concentration, floods, droughts, and all other extreme weather events [28].

The aforementioned conditions have recently been causing major problems in all areas of agriculture, as well as in the monitoring and control of insect pests [7]. Pest insects are strongly affected by climate change (Figure 1). Temperature fluctuations directly affect their biology and ecology, including reproduction, population dynamics, distribution, survival, and their relationships with the environment and natural enemies (Figure 1) [29–32]. Long-term data on insect phenology show that the occurrence of insect pests varies under changing climatic conditions [33–35]. Climate change leads to increased reproductive rate, which results in multivolatility of many insect pests (Figure 1) and consequently more crop damage [33].

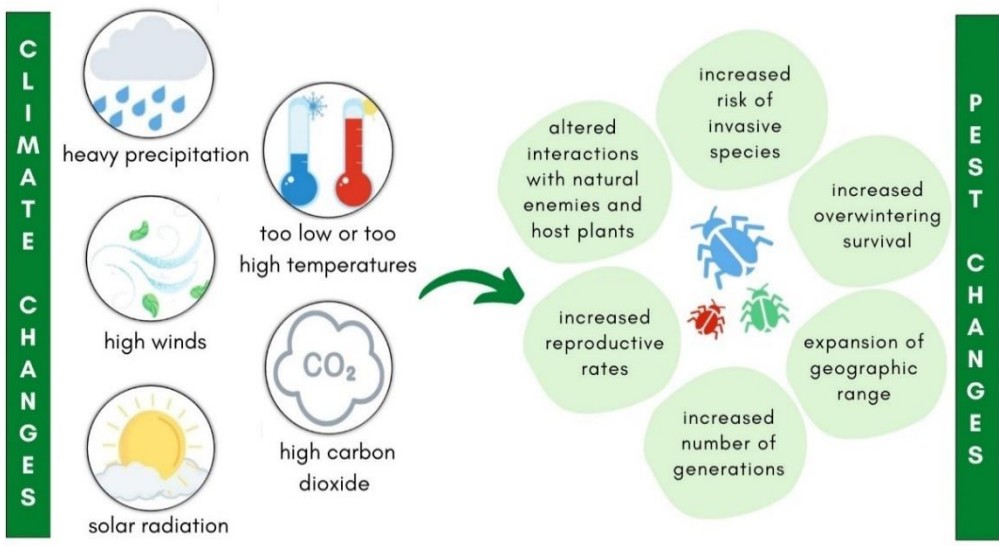

**Figure 1.** The impact of climate changes on insect pest phenology.

Altermatt [36] reported that the number of annual generations of many Central European Lepidoptera species has increased since the 1980s. Changing climatic conditions make the pests more unpredictable and their range larger (Figure 1) [37]. This also has a direct impact on the spread and establishment of invasive insect pests in new areas (Figure 1) [38]. Recent examples of rapid spread of invasive insects in Europe are *Drosophila suzukii* (Matsumura, 1931) (Diptera: Drosophilidae) [39] and *Halyomorpha halys* Stål, 1855 (Hemiptera: Pentatomidae) [40].

Apart from invasive species, climate change has a direct impact on the spread of non-invasive insect pests, for example from other continents to Europe and from warmer European regions to the north [41]. The increasing ability of insects to quickly adapt to the above conditions is becoming even more important [42], as the areas infested by plant pests and the extent of damage have seriously increased [43]. Therefore, it is crucial to adjust pest monitoring, considering that the conditions of their occurrence may change occasionally.

IPM strategies have been developed to reduce negative environmental impacts while maximizing crop yields and economic savings [44]. Among other things, this approach focuses on making decisions based on understanding how many insect pests can be tolerated in specific growing phase before economic yield losses occur (intervention thresholds). However, this approach is not always practical or possible, and when decision support systems are not available, the use of the intervention thresholds is neglected [42].

Therefore, it has recently been predicted that these thoroughly developed strategies will need to be modified to respond to the important changing climatic conditions [7,42,45]. Recognizing that climate adaptation requires widespread and long-lasting changes [46], recent attention has focused on developing new solutions for pest management [7] and monitoring.

One of the new monitoring solutions that has been increasingly used is automatic pest monitoring. Faria et al. [47], for example, emphasized the worrying impact of climate change on unexpected pest outbreaks. Pest monitoring in vineyards is currently done by traditional traps with visual inspection by growers. Recognizing that this is tedious and

low-impact work, the authors suggest capturing trap images via smartphone and remote monitoring by taxonomists to better assess unexpected outbreaks caused by climate change.

Another way to respond in a timely manner to all the unpredictability caused by climate change, including the emergence of insect pests and the establishment of invasive alien species in new geographic regions, is real-time monitoring with electronic traps (e-traps). The entomologist or farmer can check the situation in real time from the photos taken, without having to go to the field, and determine the presence (and abundance) of insects in the trap [42,48].

By monitoring climate and insect pest dynamics, farmers can adopt certain practices to respond to climate change challenges [49,50]. Modeling pest risk along with the responses of its plant hosts to climate change can also increase the ability to predict pest infestations [35]. Dong et al. [43] developed an automated system that integrates meteorology, ecology, entomology, and many other fields, as well as cutting-edge research in pest modeling, to support decision making in sustainable pest management.

Automated equipment and systems enable more profitable, sustainable, and efficient fruit production, which increasingly helps to reduce pest infestations and increase product quality and food safety [51]. Therefore, the introduction of automated technologies in traditional cropping systems is an innovative and useful solution to counter negative trends due to changing climatic conditions

## 3. Automatic Pest Monitoring Systems

Data-driven agriculture, with the help of robotic solutions incorporating AI techniques, sets the grounds for the sustainable and modern agriculture [52]. The development of innovative devices for automated monitoring has enabled end users to monitor target pest species easily and accurately [48].

There are many benefits to using automated pest monitoring equipment (Figure 2). Excessive use of pesticides and time spent on hard-to-reach work decreases, as do daily trips to inaccessible orchards [53–55]. However, depending on the pest control method, the pheromones and adhesive pads may need to be replaced occasionally to perform any maintenance [54]. Nevertheless, manual counting of pests and setting traps in orchards are operations that are not required in automatic pest monitoring systems [56,57]. Therefore, the reduction in the number of field trips leads to a significant reduction in fuel consumption and thus a reduction in carbon dioxide emissions (Figure 2) [18].

The introduction of automatic pest monitoring systems in horticultural production will ensure site-specific and environmentally friendly crop protection and thus an end product (apple fruit) with less pesticide residues.

Modern agriculture is facing tremendous technological change through the use of drones, remote sensing, intelligent decision support systems, the IoT, automated traps, and many other products of technological progress [58].

According to Potamitis et al. [59], there are several options for automatic pest control in traps: Photo-interruption is one of them, which is characteristic for pitfall or funnel traps. Detection occurs when the pest enters or falls into the trap. The way "photo-interruption" works is that the entrance of the trap is covered by a light sheet made of photodiodes and low-power infrared emitters. When the insect enters the trap, the light is interrupted, which ultimately signals the count of that pest. There is also a method of identifying insects by analyzing their wing beats in the trap.

This method of identifying pests is most commonly used in McPhail and mosquito traps. The wing beat of the incoming insect alters the flow of light, which is recorded, creating a biometric signature of a particular species [60]. Finally, the most widely used method for detecting stored-product pests is based on the vibrations generated by their movements and feeding (biting and chewing processes) [61]. However, Goldshtein et al. [62] state that imaging the trapped insects and detecting insect entry through a passage are two approaches that dominate in the development of automatic pest monitoring devices.

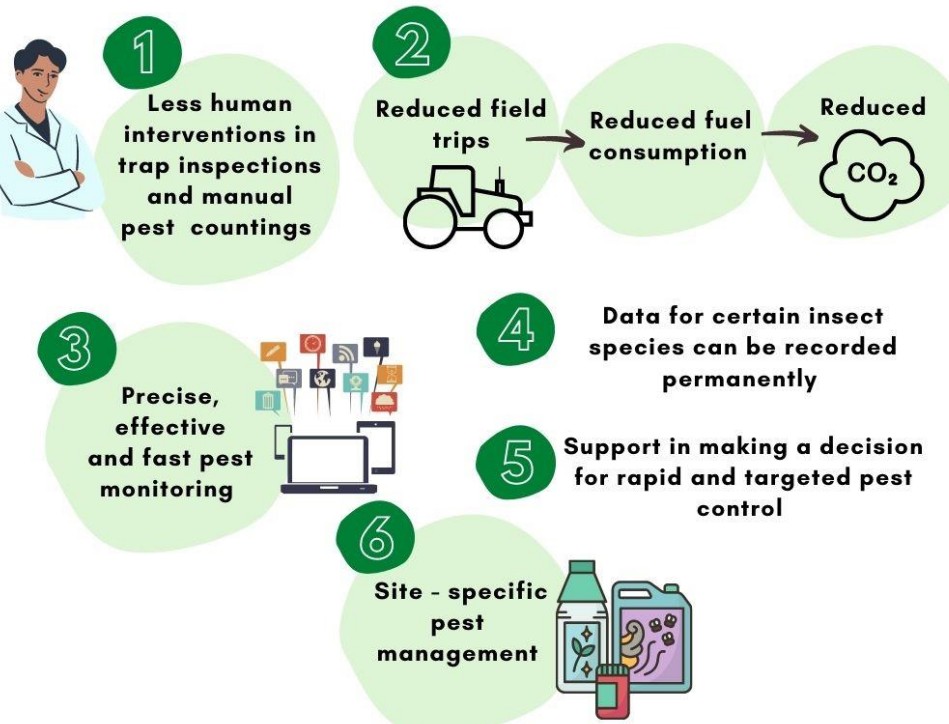

**Figure 2.** Major advantages of automatic pest monitoring systems.

Artificial neural networks (ANN-s), although significant since 1990, are now most commonly used in the development of digital systems [63]. In the development and design of a digital system, the main focus is on the image quality (brightness, resolution, focus, and the contrast of the background and the object of observation) in order to identify the observed pests as accurately as possible [64,65]. A common problem in the development of the devices is the foreign objects (leaves, branches, etc.) in the traps; therefore, the detection method should be adapted to the particular environment as much as possible [65].

Holguin et al. [56] stated that more attention should be paid to trap design. Color, size, shape, and many other parameters have the purpose of attracting pests while reducing the attractiveness to non-target pests. In particular, the development of sensors to detect different pests that can clearly distinguish non-target pests from target pests, as well as target species from each other, is encouraged. In addition, an important feature for trap efficiency is minimal energy consumption, infrequent battery replacement during monitoring, and the use of statistical and machine learning methods [66].

Monitoring systems can be classified as fully or semi-automatic. A fully automatic system is equipped with software for species identification of the trapped insects, while a semi-automatic system is based on remote identification and counting of the trapped insects by a human expert who views the images captured by the trap equipped with a camera [48]. The most effective method for pest detection is based on image analytical systems in the framework of machine learning [67–70].

The use of artificial intelligence (AI) to analyze images is a practical solution to obtain fast and accurate results [67]. The automatic pest identification system integrates multiple image processing tools to capture the geometry, morphology, and texture of photos. The processing of captured photos and videos is done by methods of analysis and manipulation of graphical components [71]. Scientists stated that this method requires a large amount of data (75,000 photos) for the proper classification of organisms [72].

However, according to Ding and Taylor [73], pest detection using a camera in a pheromone trap often encounters several challenges, including low quality photos, power consumption, and increased photo processing costs, as well as environmental factors

and pest occurrence. Regardless the commercial availability and its great potential for pest monitoring, these devices are still inaccessible to "small" farmers, due to the high cost [64,74]. Such monitoring tools are more suitable for use on regional or national level for plant protection services.

There is a potential perspective to interconnect traps among sites and create a network at local, regional, continental, or global scales, obtaining re-al-time area-wide information on insect pest infestations [59]. Therefore, in future studies, more affordable solutions should be introduced so that individual growers can also use the benefits of automatic pest monitoring. Despite all the advantages, the crucial limitation of devices used for automated pest monitoring is the availability and level of automatic pest identification and count. Several commercialized devices require a manual identification of the species or manual validation, which leaves room for further improvements in fully automated pest detection systems [20].

Automated pest monitoring is the beginning of a new scientific era in integrated pest management due to the rapid development of technologies in AI. This approach improves insect pest monitoring and early warning by integrating IoT, AI, and other advanced information technologies [75]. As the development steadily progresses, automated monitoring could be used in numerous horticultural systems and other systems for various pest species around the world. The ability to remotely detect the occurrence of insect pests and create digital records of their population dynamics, both spatially and temporally, will provide users with an immensely powerful tool to address the aforementioned insect pest monitoring challenges [76].

## 4. Automatic Monitoring of Apple Pests

### 4.1. Codling Moth (Cydia pomonella Linnaeus, 1758) (Lepidoptera: Tortricidae)

The codling moth is the most important economic and common pest of apples worldwide. Since it feeds on the fruit, targeted control measures must be implemented [77,78]. In fruit production, the most important requirement for the market is the production of high-quality fruits that do not show symptoms caused by this pest [79]. Although environmentally friendly IPM strategies, such as mating disruption, attract–kill strategy, and sterile insect technique, have been conducted, most growers rely on insecticides [80,81]. Although, 70% of insecticide treatments in apple orchards are used to control *C. pomonella*. Consequently, this pest has developed resistance to several chemical groups of insecticides [81–83].

*Cydia pomonella* develops one to four generations per year, depending on the growing area and climatic conditions [80,84]. Due to changing climatic conditions, shifts in phenology, including an increase in the number of generations, viability, and unexpected pest outbreaks of *C. pomonella* have been noted [84–86]. Pajač et al. [87] confirmed that *C. pomonella* develops an additional, third generation in Croatia in years when the sum of degree days is above average.

Considering the above, there are many important reasons for using more sophisticated, precise, and rapid techniques for early monitoring of *C. pomonella*. The first significant work on the development of sensors for automatic monitoring of *C. pomonella* was presented by Holguin et al. [56]. They proposed two electronic traps, one based on light dependent resistor (LDR) sensors and the other on infrared (IR) sensors, to detect pests when they enter the pheromone-equipped trap. These prototypes were tested in apple orchards and under laboratory conditions. By improving various aspects, these electronic traps could follow the principle of integrated pest management in precision agriculture.

Due to the need for more frequent monitoring, Guarnieri et al. [88] developed a prototype electronic trap made by modifying a commercial pheromone trap, Pomotrap, used in Italy to monitor *C. pomonella*. The system was designed to wirelessly send photos of detected insects from an orchard to a remote server. The photos were taken with a cell phone. The traps send a photo every day, which allows for a more accurate choice of timing for control methods. At the same time, modifications on the trap appearance prevented the influence of environmental factors on the correct identification of *C. pomonella*. This modified trap achieved up to 100% efficacy compared to local visual inspection.

Ding and Taylor [73] developed an automated pest monitoring system based on deep learning. This system counts and determines individuals of *C. pomonella* based on images taken inside the trap. A convolutional neural network was used to improve the photo quality. These corrections provided a better ability to distinguish insects from unwanted objects. External factors were a particular problem, causing blurring and reduced visual texture of the photos. The varying positions and sizes of the pests also made detection difficult; however, due to the convolutional layer within the network system, detection was successful.

Compared to previous attempts at pest detection, this approach does not use a pest-specific technique, and thus it can be adapted to other species and environments with minimal human effort. In real-time monitoring of *C. pomonella* in orchards, the pose variety problem often arises. Therefore, Wen et al. [89] developed a pose estimation-dependent method to identify field moth species using a deep learning system. The authors used a combination of shape, color, texture, and numerical features extracted for moth description. Later, a pyramidal stacked denoising auto-encoder (IpSDAE) was proposed to generate a deep neural network for moth detection. This model achieved a detection efficiency of 96.9%, showing that this method is suitable for automatic moth detection.

Albanese et al. [90] presented a smart trap for monitoring *C. pomonella*. The authors implemented sophisticated machine learning algorithms so that the smart trap is able to detect pests in orchards in a very short time without the need for cloud infrastructure, which is common in machine learning applications. All computations are performed on-the-node, limiting the large amount of data to a simple message of a few bytes. This solution opens many new possibilities for pest monitoring, including the optimized use of the limited energy available in the field and, consequently, the possibility of using the energy for an unlimited lifetime of smart traps. This result shows that it is possible to automate pest monitoring indefinitely, without the need for farmer intervention.

Preti et al. [54] report the development of a prototype smart trap using the latest technologies. Detection of *C. pomonella* individuals was based on preliminary analyzes of photographs taken daily under field conditions. A brand-new detection algorithm was developed to identify the trapped insects and qualitative identification parameters, such as accuracy, precision, and sensitivity, were considered for this prototype.

However, this prototype requires further improvements in both optimization of the data transmission related to power autonomy to ensure full operability throughout the monitoring season and in the refinement of the automatic detection algorithm to enable reliable machine-based count data delivery. In this case, it was demonstrated that the smart trap system could consistently deliver higher temporal resolution of capture data compared to standard monitoring at a slightly higher cost to the monitoring.

In the same year, Preti et al. [76] evaluated a commercial smart trap for monitoring the population of *C. pomonella* (Trapview; EFOS d.o.o., Slovenia) in Italy. The trap consists of pest detection software, a pheromone adhesive pad, a camera, and a solar panel. The smart trap has proven reliable in terms of data transmission speed, photo quality and battery life. The main drawback is the incorrect detection of multiple morphologically similar insects. Therefore, it is important to improve the algorithm for detecting the target pest, especially when less selective pheromones are used. The presence of different objects (insect remains, seeds, plant parts, etc.) caused interference in pest detection. Further modifications of the trap could also be related to the automatic differentiation of the sexes of *C. pomonella* species.

In addition, monitoring of the *C. pomonella* population was performed using the "Trapview" system by Pajač Živković et al. [18]. This research confirmed that this trap provides reliable pest monitoring and that there are no statistically significant differences in the effectiveness of pest monitoring compared to the classic Delta pheromone trap. Therefore, based on the aforementioned studies, it can be assumed that the classic Delta trap can be completely replaced by the smart trap in the monitoring of *C. pomonella*, as the correct intervention thresholds have been established during the growing season.

As a result, the industry is moving toward smart solutions, including remote moni-toring traps (RMT). For example, Schrader et al. [64] presented the low-cost remote monitoring

trap consisting of a plug-in imaging system in a classic Delta pheromone trap designed for monitoring *C. pomonella*. This system consists of an RGB imaging sensor combined with a microcontroller unit and associated hardware for optimized power usage and data capture.

The advantage of this system is the ability to activate sleep mode to save battery power. The system is capable of taking a picture every day. This facilitates monitoring of the *C. pomonella* population, which is one of the goals of precision agriculture devices. In addition, the construction cost of the system is about 33 US$ per unit and it has the potential to be extended to commercial applications through the Internet of Things. The plug-in imaging system developed can also be integrated with other traps for remote monitoring.

All commercially available smart traps for monitoring *C. pomonella* are summarized in Table 1.

**Table 1.** Commercially available devices for automatic monitoring of codling moth (*C. pomonella*).

| Pest | Trap | Website |
|---|---|---|
| *Cydia pomonella* (Linnaeus, 1758) | TrapView (Slovenia) | https://trapview.com/ (accessed on 15 April 2022) |
| | SightTrap™ (USA) | https://www.insectslimited.com/ (accessed on 15 April 2022) |
| | DTN Smart Trap (USA) | https://www.dtn.com/ (accessed on 15 April 2022) |
| | iSCOUT® PHEROMONE (Austria) | https://metos.at/ (accessed on 15 April 2022) |
| | Semios trap (Canada) | https://semios.com/ (accessed on 15 April 2022) |
| | CropVue trap (Canada) | https://www.cropvue.com/ (accessed on 15 April 2022) |

*4.2. Fruit Flies (Tephritidae, Drosophilidae)*

Fruit flies are one of the most destructive and economically important pests in horticulture, including apple production [91–94], which significantly affects access to the world market [95]. Fruit flies from the family Tephritidae or "true fruit flies" are highly invasive species [96]. There is a well-documented history of invasions of these species around the world, and they continue to spread rapidly despite major efforts to control their spread [97]. Invasions of fruit flies in the wake of climate change pose an increasing challenge to global food security and complicate management [98].

Species in the Drosophilidae family are referred to as "small fruit flies" or "vinegar flies" [99]. These species cause extreme losses in fruit yields and quality, while their invasion has been observed worldwide despite various quarantine procedures [97,100]. Due to changing climatic conditions, the invasive species *D. suzukii* is spreading rapidly and establishing itself in new areas around the world [101]. Reyes and Lira-Noriega [102] predicted a significant expansion of the potential range in most areas of the Northern Hemisphere by 2050, due to the increase in temperature in this area and the invasive nature of this pest.

Control of this economically important pest can be accomplished through the use of pesticides, mass trapping with pheromone baits or colored sticky traps, and the technique of releasing sterile males (SIT) [103–107]. Due to the economic importance but low intervention thresholds for these pests [108] and the already frequent insecticide treatments in apples [109], the early detection of pests is critical. As climate change further favors the invasiveness of these pests and their spread to new areas, automatic pest monitoring systems are a necessity and the best solution for more efficient and early pest monitoring under the above conditions.

There are two principles for developing automated fruit flies monitoring devices: photographing the trapped insect and sensor detection of the insect's entry into the trap. Imaging traps capture images of an insect on the trap surface and then send them to the server. They are usually designed to provide daily data by capturing and sending images of panels of trapped insects [62]. Shaked et al. [110] developed two types of electronic traps for monitoring fruit flies. The first trap contains specific volatiles for male and female *Ceratitis capitata* (Wiedemann), and the second is based on attracting *Dacus ciliatus* Loew, 1862; *Rhagoletis cerasi* (Linnaeus, 1758), and *Bactrocera oleae* (Rossi, 1790) by yellow color.

After the fruit flies are attracted, they become glued to the adhesive plate in both types of e-traps. Real-time images of the sticky plate surfaces are automatically captured and transmitted to a server. They demonstrated a remarkable ability to transmit real-time images of fruit flies and a high specificity in capturing fruit fly species. The ability of the entomologist to correctly classify fruit flies based on the images was >88%. This semi-automatic monitoring system opens the way for monitoring different fruit fly species in other crops while minimizing labor.

In addition, Roosjen et al. [111] demonstrated the feasibility of computer vision-based monitoring of *D. suzukii* using sticky traps and data from a static camera, as well as data from the same camera mounted on a flying unmanned aerial vehicle (UAV). Both data sets were taken outdoors under different conditions. Using an image database of 4753 annotated flies, they trained a ResNet-18-based deep convolutional neural network to recognize each individual and distinguish males from females. In contrast to previous work, this paper presented a fully automated system highlighting the use of drones to monitor fruit flies in various fruit crops.

On the other hand, sensor-based traps are designed to provide a time stamp for every entry of insect pest [62]. The same author, Goldshtein et al. [62], developed an automatic trap for detection of *C. capitata*. The Medfly-AT trap is cylindrical in shape and contains optical sensors to detect pest individuals. Field tests were conducted to evaluate the efficiency of the automatic trap, and the accuracy ranged from 88% to 100%. Although the trap shows promise in reducing insecticide use, extensive field testing is needed to implement it.

The frequency of insect wing beating can also be used for identification, as it depends on the physiological characteristics of each fruit fly species [112]. Therefore, Potamitis et al. [59] developed an electronic trap as a modification of the classical trap for fruit flies (McPhail trap).

The insects are identified using the high-quality recordings of the frequency content of their wingbeat, with an accuracy of 91%. Species identification can be done exclusively in situ with this version of the trap. With in situ identification, the trap can only distinguish fruit flies from completely different insects, or it cannot distinguish fruit flies at the species level (e.g., the difference between *Bactrocera dorsalis* (Hendel, 1912) and *C. capitata*). To improve this trap, Potamitis et al. [113] presented a novel bimodal optoelectronic sensor based on a Frensel lens and a stereo recorder that records the wingbeat of an insect in flight and wirelessly reports the count and species identity.

They incorporated some of this technology into an electronic trap for fruit flies mentioned earlier. Unlike the systems mentioned above, this type of system can distinguish *C. capitata* and *B. oleae* with 98.99% accuracy. It is also important to emphasize that the said electronic trap is optimized in terms of detection accuracy and power consumption, in addition to being affordable for the end user. In addition, Sandrini Moraes et al. [114] developed an optoelectronic sensor that detects the signal of partial masking of infrared light by the flies' wing beat.

They used the aforementioned sensor in a classical McPhail trap to identify *C. capitata* and *Anastrepha fraterculus* (Wiedemann) in real time, and both species were detected with 95% accuracy. This electronic trap, like those mentioned above, can be integrated into an automated warning system to inform farmers of fruit fly infestations and can contribute to precision agriculture. Commercial electronic traps that can be used to monitor different fruit fly species in apple production are listed in Table 2.

In order to obtain very accurate prediction results and early detection of pests, it is necessary to consider potential factors that influence pest population dynamics, such as biological characteristics and physical environmental parameters [115]. For this reason, Jiang et al. [15] developed a remote pest monitoring system or an automatic trapping and counting device that is a modification of the traditional fruit fly trapping tube and includes an automatic counting module that is inserted into the tube. It records captured *B. dorsalis* individuals, recognizes them by their biological characteristics, and then sends the information to the remote monitoring platform (RMP).

**Table 2.** Commercially available devices for automatic monitoring of fruit flies (Tephritidae and Drosophilidae).

| Category | Pest | Trap | Website |
|---|---|---|---|
| Fruit flies | *Drosophila suzukii* (Matsumura, 1931) | iSCOUT® FRUIT FLY (Austria) | https://metos.at/ (accessed on 26 April 2022) |
| | | TrapView (Slovenia) | https://trapview.com/ (accessed on 26 April 2022) |
| | *Ceratitis capitata* (Wiedemann, 1824) | iSCOUT® FRUIT FLY (Austria) | https://metos.at/ (accessed at 15 April 2022) |
| | | TrapView (Slovenia) | https://trapview.com/ (accessed on 15 April 2022) |
| | *Rhagoletis pomonella* (Walsh, 1867) | iSCOUT® COLOR TRAP (Austria) | https://metos.at/ (accessed on 26 April 2022) |
| | | RapidAIM (Australija) | https://rapidaim.io/ (accessed on 26 April 2022) |

The RMP collects environmental data and the number of flies captured, and then sends all data to the Host Control Platform (HCP) via the wireless Global System of Mobile Communication (GSM). The function of the HCP is to receive, store, display, and analyze the database online and provide early warnings. The authors combined the traditional trapping method with modern communication technology to provide real-time information on field conditions and pest population dynamics.

Okuyama et al. [116] used this automated pest monitoring system with excellent networks and a large amount of real-time data to study the daily count data of *B. dorsalis* population dynamics. Later, the monitoring system was enhanced with wireless sensor network (WSN) technologies and is now capable of monitoring microclimate, meteorological and pest data in the field [117].

Liao et al. [118] wanted to improve the above type of system and develop a more accurate device. Therefore, they designed a monitoring system based on two different wireless protocols: GSM and ZigBee, with three main components: Remote Sensing Information Gateway (RSIG), a Host Control Platform (HCP), and Wireless Monitoring Nodes (WMNs). The WMNs transmit the collected data (relative humidity, illumination, temperature, and number of *B. dorsalis* individuals captured) to the RSIG, and the RSIG forwards the data to the database server (HCP) for storage and analysis.

The server can process the data and classify the information into three event types after analysis: a normal status event, a pest outbreak event, and a sensor failure event, which can be accessed through an online platform. This early warning system can be easily deployed in different orchards without the need for additional manpower due to its machine learning techniques and receiving alerts on cell phones. This early warning system has great potential to help farmers monitor fruit flies in their orchards.

In addition, Chuang et al. [119] proposed a model that provides a 7-day prediction of the population dynamics of the aforementioned pest *B. dorsalis*. More specifically, it provides unique datasets on *B. dorsalis* population dynamics using WSN technology. Moreover, the distribution and biology of this fly will change due to global climate change. Therefore, it is important to build new predictive models by reconsidering the ecological behavior of *B. dorsalis* so that apple production and production of other crops can be improved and farmers will be able to respond quickly to potential outbreaks of the pest.

Since previous studies mostly used long-term data to predict pest populations [119] and such data are not useful in all cases, Jiang et al. [120] therefore proposed an interval type-2 fuzzy logic system (IT2FLS) based on short-term data to predict the population dynamics of *B. dorsalis*. Two models were developed and integrated into the proposed IT2FLS. Compared to previous population dynamics prediction models, these models shorten the training interval and provide more accurate results.

Zhong et al. [121] developed a visual counting and detection system for six insect species, including fruit flies. In their approach, a camera was installed at an optimized position to monitor yellow sticky traps. The insects were detected from the captured images and counted using the "YOLO" algorithm and Support Vector Machines. In addition, insect counting, and detection was implemented on a Raspberry Pi system.

The average counting accuracy was 92.50%, and the average classification accuracy was 90.18%. The proposed system is quite easy to use and at the same time provides accurate detection data or monitors the spread of insects in real time. This system can be combined with the various environmental information to form an integrated service platform that can predict the population dynamics and probability of occurrence at an early stage to take appropriate control measures.

The damage caused by economically important fruit flies could also be significantly reduced by using automatic or semi-automatic image analysis systems [22]. Due to the shortcomings of conventional fruit fly classification systems, Peng et al. [122] proposed a convolutional neural network algorithm that automatically extracts features to build a classification model for the four *Bactrocera* species. The developed model automatically extracts the features of the fruit fly pests for effective identification with an accuracy rate of 97.19% and solves the problems caused by the manual classification methods.

In addition, Wang et al. [22] developed an image identification system for fruit flies called AFIS1.0, which combines automatic image identification and manual interactive methods based on image queries with a user-friendly interface. The system works in such a way that the user only needs to input images, select feature areas, and click the button to get the image identification results. At this stage, AFIS1.0 can be used without specific knowledge of the Tephritidae family species. This brings the application of computer vision technology to detect economically important fruit flies within reach for farmers. However, the authors stated that the software will certainly be improved with new features in the future.

Leonardo et al. [123] also developed an automatic and semi-automatic system for morphological recognition of fruit fly species of the genus *Anastrepha* Schiner, 1868 using image processing and machine learning techniques. This kind of system, as with those above, can help experts reduce the time spent on lengthy insect analyzes and the ecological losses caused by these fruit fly species. They used mid-level image representations based on local descriptors to identify three species of the genus *Anastrepha*. The authors explored local image descriptors based on key points and machine learning techniques to facilitate detection of these pests.

Their approaches achieved excellent results in terms of effectiveness compared to the state-of-art techniques. In addition, Faria et al. [124] developed a system to identify different species of the genus *Anastrepha* based on image datasets of the wings and the aculeus, an ovipositor structure. They used a multimodal fusion classifier approach and an image analysis system, respectively, to automatically recognize and distinguish three species: *Anastrepha fraterculus* (Wiedemann, 1830), *Anastrepha obliqua* (Macquart, 1835), and *Anastrepha sororcula* Zucchi, 1979, based on the features of the aforementioned structures.

In these experiments, the multimodal approach of Fuzzy Support Vector Machines (FSVM) achieved a classification accuracy of 98.8% under laboratory conditions. This means that the automatic identification of these species based on image analysis and learning techniques is an effective alternative to the conventional tedious and inaccurate methods currently used.

In contrast, Blasco et al. [125] developed an automatic system for distinguishing males and females of *C. capitata* species, consisting of a backlight system and image processing algorithms. The determination is made using five high-resolution images of each insect. The program analyzes the contour of the abdomen to detect the presence of the ovipositor, as well as the characteristic spatulate setae of males, with an error rate of 0.6% for females and 0% for males.

### 4.3. Other Important Apple Pests

#### 4.3.1. Pear Leaf Blister Moth (*Leucoptera maifoliella* (O. Costa, 1836)) (Lepidoptera: Lyonetiidae)

The pear leaf blister moth is an economically important pest in apple production [126,127]. It is a multivoltine species [128], and due to changing climatic conditions, it is becoming more common and with larger populations [129,130].

The finding that a complete generation of this pest develops successively in a short period of time after overwintering can be used for its management [77]. The success of management is determined by two remarkable events: (a) monitoring of the flight of the first generation and the beginning of oviposition and (b) embryonic development of caterpillars, their perforation in the leaf and the initial development of mines up to 2 mm in diameter [130].

Synthetic pheromones for monitoring *L. malifoliella* have proven useful [129]. However, the damage caused by this pest in apple orchards occurs because the overwintering developmental stage is noticed too late due to the small size of the insects and the hiding behavior of the larvae [130,131]. Late measures of targeted chemical protection led to worse results [130]. Therefore, the use of automated systems for monitoring this pest enables its early detection and timely and effective application of insecticides.

Grünig et al. [132] presented a system for monitoring *L. malifoliella* in apple orchards based on Big Data and deep-learning algorithms. The authors used 52,322 photographs taken under field and standard conditions. Deep neural networks (DNNs) were used to classify damage to apple leaves and examine the phenology of seven standard classes of damage (Undamaged, Mines/Blister Moth Detected, Physical Damage, Lepidoptera, Brown Spot, Powdery Mildew, and Feeding Damage from phytophagous insects) predicted by the DNNs.

They also linked predicted damage occurrence with meteorological data to model damage phenology. They proposed to solve this problem with data collection by citizen science or drones. This work also opens new possibilities for early warning of other economic pests in apple production.

### 4.3.2. Brown Marmorated Stink Bug (*Halyomorpha halys* Stål, 1855)

The brown marmorated stink bug has recently received much attention in many scientific studies [133,134]. This pest is extremely harmful and polyphagous. It feeds on a wide range of hosts (more than 300 plant species) [135,136] including economically important crops and also apple [137–139]. It can cause 100% crop losses in fruit and corn crops [136,140]. As populations of *H. halys* increased significantly in the U.S., it very soon replaced lepidopteran pests (*C. pomonella* and *Grapholita molesta* (Busck, 1916)) (Lepidoptera: Tortricidae) as major pests in orchards [138]. Early damage is manifested by fruit deformation, and later feeding by *H. halys* usually leads to the formation of necrotic areas and eventually to fruit flesh disintegration [136].

Control of this pest is quite difficult due to its high mobility and polyphagy [140,141]. First, the use of insecticides can be very harmful to beneficial arthropods and lead to an increase in pest outbreaks. Furthermore, due to the genetic structure of the pest, there is a possibility that it will respond to this extensive chemical control with resistance [142].

In addition, this pest is highly invasive [136,142], and its life cycle is strongly influenced by changing climatic conditions, further complicating the management of this species in terms of both geographic distribution and population growth [143]. Indeed, Kistner et al. [143] suggested that the number of *H. halys* generations may be increasing due to climate change, causing the species to become multivoltine in the northern latitudes of North America and Europe, where it is currently considered univoltine. This means that key horticultural areas in Europe, the northeastern United States, and southeastern Canada are most at risk from this invasive pest. Therefore, the need for rapid, accurate, and real-time monitoring of this invasive species can only be met through the use of automated pest monitoring systems.

Considering the unstoppable spread of this pest [136] and all the problems mentioned above, efficient, and sustainable monitoring strategies are increasingly being investigated. For example, models capable of predicting *H. halys* population dynamics based on degree days. These models improve the prediction of future population trends of this pest and ultimately contribute to efficient management [144].

However, Lippi et al. [145] developed a pest detection system to identify true bugs (*H. halys*, *Gonocerus acuteangulatus* (Goeze, 1778) (Hemiptera: Coreidae), and *Palomena*

*prasina* (Linnaeus, 1761)) (Hemiptera: Pentatomidae) on white sticky traps in a cereal field. They used the YOLOv4 (You Only Look Once) convolutional neural network (CNN) model based on deep learning.

It was trained on a user-defined dataset collected in an orchard under realistic conditions and achieved an average accuracy of 94.5%. Moreover, the real-time performance was experimentally verified on a built-in system (NVIDIA Jetson Xavier) that can be easily installed on any mobile platform. The research was conducted in hazelnut production but should be applicable to pest detection in other crops, especially apples.

### 4.3.3. Oriental Fruit Moth (*Grapholita molesta* Busck, 1916)

Oriental fruit moth is primarily a pest of stone fruits (peaches, nectarines, and apricots); however, it also causes significant damage to apples [146,147]. Initially, it causes wilting and withering of shoots, and later in the growing season, the fruits become wormy and lose their organoleptic properties [128,147]. This pest is active over a period of four months and is significantly more numerous than *C. pomonella* [148].

Due to changing climatic conditions, increasing areas are suitable for invasion by this pest, suggesting that constant monitoring is needed to respond quickly and reduce the potential spread of *G. molesta* in the main commercial horticultural areas [149].

Therefore, an effective and accurate method for monitoring can be found in automatic pest monitoring systems [150]. Ascolese et al. [151] monitored the population of *G. molesta* using commercially available electronic traps (iMETOS iSCOUT® pheromone model and Trapview Standard model) in peach orchards. These electronic traps consisted of a solar panel, a housing trap, an adhesive pad with pheromone, and a camera with SD memory card that sent images to a remote server. The Trapview Standard model had a problem with detecting other objects (leaves and twigs) from trapped insects. However, the dimensions and color of the trap proved effective in capturing moths.

On the other hand, iMETOS iSCOUT® also had a problem; however, this was with sending photos to the platform for further analysis, which was due to a weak network signal. In addition, Pérez-Aparicio et al. [150] developed an affordable smart trap for monitoring *G. molesta*. It consisted of a Raspberry Pi system with an infrared camera powered by open-source software and housed in a plastic box. Images were downloaded to a computer via WiFi from the Raspberry's SD card. The traps were reliable and easy to use; therefore, with further improvements, traditional pest monitoring methods can be replaced by the remotely controlled devices for monitoring *G. molesta* [151].

Commercial electronic traps that can be used to monitor various moth, true bug, thrips, whitefly, and wasp pests in apple orchards are summarized in Table 3.

**Table 3.** Commercially available devices for the automatic monitoring of other important apple pests.

| Category | Pest | Trap | Website |
|---|---|---|---|
| Moths | *Adoxophyes orana* (Fischer Röslerstamm, 1834) | iSCOUT® PHEROMONE (Austria) | https://metos.at/ (accessed on 15 April 2022) |
| | *Synanthedon myopaeformis* (Borkhausen, 1789) | iSCOUT® PHEROMONE (Austria) | https://metos.at/ (accessed at 15 April 2022) |
| | *Zeuzera pyrina* (Linnaeus, 1761) | iSCOUT® PHEROMONE (Austria) | https://metos.at/ (accessed at 15 April 2022) |
| | *Pammene rhediella* (Clerck, 1759) | iSCOUT® PHEROMONE (Austria) | https://metos.at/ (accessed at 15 April 2022) |
| | *Choristoneura rosaceana* (Harris, 1841) | TrapView (Slovenia) | https://trapview.com/ (accessed at 15 April 2022) |
| | | Semios trap (Canada) | https://semios.com/ (accessed at 15 April 2022) |
| | *Cydia molesta* (Busck. 1916) | TrapView (Slovenia) | https://trapview.com/ (accessed at 15 April 2022) |
| | | Semios trap (Canada) | https://semios.com/ (accessed at 15 April 2022) |

**Table 3.** *Cont.*

| Category | Pest | Trap | Website |
|---|---|---|---|
| True bugs | *Halyomorpha halys* (Stål, 1855) | iSCOUT® BUG (Austria) | https://metos.at/ (accessed at 15 April 2022) |
| Thrips | *Frankliniella occidentalis* (Pergande, 1895) | iSCOUT® COLOR TRAP (Austria) | https://metos.at/ (accessed at 15 April 2022) |
| Whiteflies | *Quadraspidiotus perniciosus* (Comstock, 1881) | iSCOUT ® PHEROMONE (Austria) | https://metos.at/ (accessed at 15 April 2022) |
| Wasps | *Hoplocampa testudinea* (Klug, 1816) | iSCOUT ® COLOR TRAP (Austria) | https://metos.at/ (accessed at 15 April 2022) |
|  | *Hoplocampa flava* (Linnaeus, 1761) | iSCOUT ® COLOR TRAP (Austria) | https://metos.at/ (accessed at 15 April 2022) |

## 5. Conclusions

Due to changing climatic conditions, there is a need to adapt the current monitoring techniques. As a result, there are increasing amounts of automated monitoring systems. In the presented work, automated pest monitoring in apple production is mainly focused on codling moth (*C. pomonella*) monitoring due to its economic importance. However, the automated techniques presented in this article for monitoring other apple pests could also be used in apple production. The listed commercial smart traps are mostly made by modifying classical traps for monitoring specific pests. Therefore, this method is the simplest and most accessible for automatic pest monitoring. Recently, the development of automated pest monitoring systems is progressing and could completely replace classical monitoring methods, adding additional improvements and better accessibility for farmers.

Automated monitoring systems could become an indispensable part of sustainable apple production, successfully improving early pest detection and control. Most importantly, these techniques and devices will help reduce the environmental footprint while improving pest monitoring in apple production under changing climatic conditions.

**Author Contributions:** Conceptualization, D.Č., I.M. and I.P.Ž.; validation, I.P.Ž., D.L. and T.K.; investigation, D.Č. and I.M.; resources I.P.Ž.; data curation, D.Č. and I.M.; writing—original draft preparation, D.Č., I.M., I.P.Ž., D.L. and T.K.; writing—review and editing, I.P.Ž., D.L. and T.K.; visualization, D.Č., I.M. and I.P.Ž.; supervision, I.P.Ž., D.L. and T.K.; project administration, I.P.Ž.; funding acquisition, I.P.Ž. All authors have read and agreed to the published version of the manuscript.

**Funding:** The publication was supported by the Open Access Publication Fund of the University of Zagreb Faculty of Agriculture and the European Regional Development Found through project AgriART comprehensive management system in the field of precision agriculture (KK.01.2.1.02.0290).

**Data Availability Statement:** No new data were created or analyzed in this study. Data sharing is not applicable to this article.

**Acknowledgments:** The authors thank the European Union, which supported the project AgriART comprehensive management system in the field of precision agriculture (KK.01.2.1.02.0290) through the European Regional Development Fund within the Operational Programme Competitiveness and Cohesion (OPCC) 2014–2020.

**Conflicts of Interest:** The authors declare no conflict of interest.

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
