# Peer review of "Automatic Pest Monitoring Systems in Apple Production under Changing Climatic Conditions"

_horticulturae, doi:10.3390/horticulturae8060520_

Round 1

Reviewer 1 Report

The article is very well written and comprehensive, addressing a very sensitive topic for the application and adoption of integrated pest management. This article could take a slightly more emphatic approach to the challenges and constraints for current use and application. Suggestion: discuss a little more about the challenges, study needs and limitations for use.

Reviewer 2 Report

This manuscript is a good contribution for the new challenges facing by producers and farmers. Overall, I have found this manuscript interesting since it put together several new monitoring techniques that can be useful for apple producers. 

There are some small details  that need to be addressed in order to improve the  quality of this manuscript. 

See attached file.
